# Quantitative Identification Method for Glass Panel Defects Using Microwave Detection Based on the CSAPSO-BP Neural Network

**DOI:** 10.3390/s23031097

**Published:** 2023-01-18

**Authors:** Jun Fang, Zhiyang Deng, Jun Tu, Xiaochun Song

**Affiliations:** 1School of Mechanical Engineering, Hubei University of Technology, Wuhan 430068, China; 2Hubei Key Laboratory of Modern Manufacturing Quality Engineering, Wuhan 430068, China

**Keywords:** glass panel, chaos, simulated annealing algorithm, particle swarm optimization algorithm, microwave detection, quantitative identification

## Abstract

To address the problem of the quantitative identification of glass panel surface defects, a new method combining the chaotic simulated annealing particle swarm algorithm (CSAPSO) and the BP neural network is proposed for the quantitative evaluation of microwave detection signals of glass panel defects. First, the parameters of the particle swarm optimization (PSO) algorithm are dynamically assigned using chaos theory to improve the global search capability of the PSO. Then, the CSAPSO-BP neural network model is constructed, and the return loss and phase of the microwave detection echo signal of glass panel defects are extracted as the input feature quantity of the network, from which the intrinsic connection between input and output is found through network training and testing to achieve the prediction of the depth and width of glass panel surface defects. The results show that the CSAPSO-BP network model can more accurately characterize the defect geometry of glass panels than the PSO-BP network model.

## 1. Introduction

The quality of a glass panel is a key factor in the touch performance and optical performance of the panel. In the glass shape processing, tempering, polishing, screen printing and other production processes, it is easy to cause panel scuffs, scratches, chipping, bubbles and other defects, among which crack-like defects are the most common and should be identified and eliminated in real time during the manufacturing and production processes [1]. At present, the surface defects of glass panels are detected mainly by manual inspection and machine vision inspection [2]. However, manual inspection is not only costly and inefficient, but can also easily lead to false detection or a missed inspection due to visual fatigue of the inspectors [3]; machine vision inspection requires different light source systems designed for different types of defects [4], and the high real-time accuracy of the defect recognition algorithm also needs to be improved. In contrast, microwave inspection technology [5] has the advantages of low power consumption, noncontact, easy operation, and no ionizing radiation, so it has a better detection effect for different types of defects on glass panels, and it is easy to achieve automated detection [6].

At present, microwave inspection technology is mostly based on qualitative detection, which can only determine the presence or absence of defects, and it is difficult to identify the shape and geometry of defects and other information. The chaotic simulated annealing particle swarm optimization algorithm takes advantage of the ability of simulated annealing to make sudden jumps in local extremes, improving the prematureness of the particle swarm optimization algorithm. Chaos theory is also introduced to improve the search capability of the algorithm [7]. However, the algorithm is currently mostly applied to solve problems such as shop scheduling, electric vehicle charging station planning, and Weibull distribution parameter estimation [8]. There are still few reports on defect detection and quantitative identification.

To this end, a quantitative defect identification method combining the BP neural network and the chaotic simulated annealing particle swarm optimization algorithm is proposed for the quantitative identification of the microwave detection of glass panel surface crack-like defects. This article is organized as follows. Section 2 introduces the basic principles of microwave inspection and establishes a finite element analysis model for the microwave inspection of a glass panel, and investigates the effect of defect size and microwave signal. Section 3 reveals the shortcomings of the current BP neural network and particle swarm optimization algorithms, and proposes a CSAPSO-BP neural network algorithm. Section 4 verifies the effectiveness and superiority of the CSAPSO-BP neural network algorithm for the microwave detection of glass panel defects and quantitative identification of defects by training and testing the CSAPSO-BP neural network model. Conclusions and future work are summarized in Section 5. 

## 2. Finite Element Analysis Model for the Microwave Inspection of a Glass Panel

### 2.1. Detection Principle

The principle of microwave detection of surface defects on glass panels is shown in Figure 1. The rectangular waveguide probe emits a certain frequency band of microwaves to the surface defects of the glass panel; on one hand, the microwaves will scatter a lot at the nonuniform interface, which weakens the energy received by the probe; on the other hand, due to the increased distance between the microwave excitation source and the panel surface caused by the defect, the phase of the echoes received by the probe will also change. Therefore, by studying the return loss and phase change in the reflected wave, it is possible to achieve glass panel surface defect detection.

### 2.2. Calculation Model

Based on the microwave detection principle, the interaction between the microwaves and the defect on the glass panel surface is calculated numerically using CST electromagnetic field simulation software. Firstly, a panel model with dimensions of 60 mm × 40 mm × 5 mm was established with parameters set to a relative magnetic permeability of $\mu_{r}$ = 1 and a complex dielectric constant of $\varepsilon_{r}$ = 5.33 − j0.096. A slotted defect parallel to the long direction (X-direction) of the panel was established in the center of the panel. To improve detection sensitivity, a rectangular waveguide with a length of 15.8 mm and a width of 7.9 mm was selected, and the lifting distance between the waveguide and the glass panel was 0.5 mm. Through simulation calculations, parameters such as return loss and reflection coefficient phase of the microwave signal are mainly obtained. The 3D simulation calculation model for the microwave detection of defects on panel surfaces is shown in Figure 2.

### 2.3. Calculation Result

The surface defect size (3 mm × 0.1 mm × 0.1 mm) of the glass plate was swept in the x-direction with a rectangular waveguide probe at 13.722 GHz. Figure 3a shows the variation curve of the microwave return loss along the direction of the probe sweep. The return loss first drops and then rises. When the probe center is 3 mm from the crack center, the return loss decreases gradually. Until the center of the probe and the center of the defect are completely coincident, the return loss stops decreasing. The whole curve has symmetry. Figure 3b shows the phase change curve along the probe scanning direction. It is not difficult to find that the whole curve has the same symmetry, and the phase first gradually decreases, and when the probe center is 6 mm away from the crack center, i.e., when the defect is completely inside the inside of waveguide port, the phase starts to increase until the center of the rectangular probe coincides exactly with the geometric center of the crack, and the phase reaches its highest value.

Figure 4 and Figure 5 show the echo signals of defects with different depths and widths, respectively. From the figures, it can be seen that the echo loss and phase at the defect center point show a certain monotonicity as the width increases. However, the effect of defect depth variation on the echo loss and phase does not show a clear correlation. Therefore, it is difficult to make an accurate quantitative evaluation of the defect geometry by simply using the linear fitting method. 

## 3. CSAPSO-BP Neural Network Algorithm

### 3.1. BP Neural Network

The BP neural network [9,10,11] is a multilayer feedforward neural network, and its topology is shown in Figure 6. The main feature of this network is that the signal is transmitted forward and the error is propagated backward. In forward transmission, the input signal is processed from the input layer through the implicit layer to the output layer. The neuron state in each layer only affects the neuron state in the next layer, and if the desired output is not obtained in the output layer, it is transferred to backpropagation, and the weights and thresholds of the network are adjusted according to the prediction error so that the predicted output of the BP neural network continuously approximates the desired output [12,13,14].

In Figure 6, n, j and m are the number of nodes in the input layer, the number of nodes in the hidden layer and the number of nodes in the output layer, respectively. $T_{1}$, $T_{2}$, …, $T_{n}$ are the input samples, $P_{1}$, …, $P_{m}$ are the output samples, $\omega_{ij}$ and $\omega_{jk}$ are the weight from the input layer to the hidden layer and the weight from the hidden layer to the output layer, respectively.

### 3.2. PSO Algorithm

The particle swarm optimization algorithm [15,16,17] is a population optimization algorithm derived from bird flock foraging, which focuses on guiding the optimization search through mutual cooperation and mutual search among flocks of birds [18,19]. The particle swarm algorithm is first initialized as a group of random particles that update themselves in iterations by tracking individual and global extremes [20]. In this process, the current velocity and position of each particle are mainly determined by Equations (1) and (2).
(1)$$\nu_{id}(t+1)=\omega \times \nu_{id}(t)+c_{1}\times r_{1}\times \left[x_{id}^{*}-x_{id}(t)\right]+c_{2}\times r_{2}\times \left[x_{gd}^{*}-x_{id}(t)\right]$$
(2)$$x_{id}(t+1)=x_{id}(t)+\nu_{id}(t+1)$$

In the formula, $c_{1}$ and $c_{2}$ are acceleration constants to regulate the maximum learning step; $r_{1}$ and $r_{2}$ are random numbers of (0, 1) to increase the randomness of the particle search; ω is the inertia weight; and $x_{gd}^{*}$ is the best position that all of the particles in the group have found so far.

Since the search performance of the PSO algorithm has a certain dependence on the parameters, in many cases, the size of the parameter values directly affects whether the algorithm converges and the accuracy of the solution results. At the same time, in the PSO algorithm, when a particle finds a local optimal solution, other particles will be attracted by the optimal solution and quickly gather in its vicinity, thus making the whole algorithm converge prematurely and fall into a local optimum [21]. Therefore, there is a need to improve the global search capability of the algorithm.

### 3.3. CSAPSO-BP Neural Network Algorithm

In order to solve the problem of the poor global search ability of the PSO algorithm, the chaos theory and simulated annealing algorithm are combined to optimize the PSO algorithm, and the chaotic simulated annealing particle swarm optimization algorithm (CSAPSO) is proposed. Chaos theory is used to dynamically assign the parameters $r_{1}$ and $r_{2}$ of the particle swarm optimization algorithm to produce a population of excellence. Combined with a simulated annealing algorithm, the algorithm is able to accept some poor solutions with a certain probability, which improves the global search ability of the algorithm itself. The BP neural network has good adaptability in function approximation, nonlinear function fitting and online prediction. However, it has the problem of slow convergence and easily falls into local minima, so the CSAPSO algorithm is used to optimize the BP neural network, the weights and thresholds of the BP neural network are used as the optimal solutions of the optimization algorithm, and the mean square error is used as the fitness function of the algorithm. The flow chart of the algorithm is shown in Figure 7. The specific algorithm flow is as follows:

Step 1: Start. Determine the number of neurons in the input layer, hidden layer and output layer of the BP neural network.

Step 2: Initialize the relevant parameters of the algorithm, i.e., the initial position, velocity, inertia weight, acceleration constant of the particles, the size of the population, initial temperature and the maximum number of iterations. Import the return loss and phase training sets and normalize them.

Step 3: Calculate the initial fitness value of the particles.

Step 4: Termination condition judgment. If the individual fitness value of the optimal solution meets the set error requirement or the number of iterations meets the requirement, the optimal weights and thresholds are outputted to the BP neural network; if not, the next step is executed.

Step 5: Compare the fitness values to determine the individual optimum and the global optimum.

Step 6: Follow Equations (1) and (2) to update the velocity and position of the particle and recalculate the fitness value of the new particle.

Step 7: Calculate the change in the fitness value caused by the two positions.

Step 8: The judgment of fitness difference Δf. If the Δf<0 or exp (−Δf/T)>rand requirement is met, the new position is accepted; otherwise, keep the old position, perform the cool-down process and return to step 4.

Step 9: The weights and thresholds determined by the optimized CSAPSO algorithm are used as the weights and thresholds of the BP neural network, and then the neural network is used for training. If the maximum training time or minimum error are reached, the depth and width of defects will be outputted. If not, keep training.

## 4. CSAPSO-BP Neural Network Training and Testing

### 4.1. Extraction of Feature Parameters

Based on the numerical calculation model shown in Figure 2, the defects with a length of 3 mm, a width of 0.1 mm and a depth range of 0.05~2 mm, and the defects with a length of 3 mm, a depth of 0.1 mm and a width range of 0.1~4 mm were numerically calculated, respectively; forty sets of data were collected for each. The return loss and phase are selected as the input of the BP neural network, and the depth and width of the cracks are used as the network outputs.

### 4.2. Training and Testing

From the above 40 groups of simulation calculation data with different depths and widths, 5 groups of data with depths of 0.05 mm, 0.5 mm, 1 mm, 1.5 mm and 2 mm and 5 groups of data with widths of 0.1 mm, 1 mm, 2 mm, 3 mm and 4 mm were selected as the test set, and the other 35 groups of data were selected as the training set. The CSAPSO-BP neural network model was established by using MATLAB. The input layer was defined as two nodes, the output layer as one node and the hidden layer as eight nodes. The learning rate of the neural network was 0.05 and the training target error was 10^−5^. After dozens of tests, the parameters of the chaotic simulated annealing particle swarm algorithm were determined as follows: the population size was 10, the number of iterations was 100, the learning factors $c_{1}$ and $c_{2}$ were 2, $\omega_{\min}$ and $\omega_{\max}$ were 0.4 and 0.9, respectively, the initial temperature was 1000 and the temperature decay coefficient was taken as 0.98. After normalizing the sample data, the PSO-BP neural network model and CSAPSO-BP neural network model were trained and tested, and finally, the predicted results were treated with reverse normalization.

### 4.3. Result Analysis

Figure 8 and Figure 9 show the test set R-coefficients and MSE curves for the prediction of defect width for the two algorithms CSAPSO-BP and PSO-BP, respectively. Figure 10 and Figure 11 show the test set R-coefficients and MSE curves for the prediction of defect depth for both the CSAPSO-BP and PSO-BP algorithms, respectively. It can be seen from the plots that the R-coefficients of the CSAPSO-BP algorithm are somewhat higher than the PSO-BP for both defect width prediction and depth prediction, and the algorithm of CSAPSO-BP converges faster.

The prediction results of the two algorithms are shown in Table 1 and Table 2. From the table, it can be seen that the CSAPSO-BP neural network is significantly better than the PSO-BP neural network in terms of the accuracy of the prediction results for defect depth and width.

Figure 12a shows the error comparison between the two algorithms for the prediction of the defect width. It can be seen that the average error percentage of the PSO-BP neural network for predicting the defect width is 3.226%, while the average error percent-age of the CSAPSO-BP neural network for predicting the defect width is 0.382%, which is smaller than that of the PSO-BP neural network. Figure 12b shows the error percentage of defect depth prediction for both algorithms. the average error of PSO-BP neural network is 8.406%, while the average error of predicted depth of CSAPSO-BP neural network is 2.784%, which is more accurate than that of PSO-BP neural network. The above results show that the CSAPSO-BP neural network has less error and higher accuracy than the PSO-BP neural network in the prediction of defect depth and width, which proves the feasibility of the CSAPSO-BP neural network for quantitative identification of surface defects.

## 5. Discussions and Conclusions

This paper combines the nondestructive microwave testing method and the CSAPSO-BP neural network to construct a quantitative network model for identifying defects on the surface of glass panels and to achieve a quantitative evaluation of the width and depth of crack-like defects on the surface of panels. The CST microwave simulation software was used to calculate the changes in microwave signals when the waveguide probe swept the glass panel surface defects, and the loss and phase of the echo signal at the center of the defect were extracted as the input feature parameters of the CSAPSO-BP neural network. The network was trained and tested, and the recognition effect was compared with that of the PSO-BP neural network. The results show that the defect geometry recognition algorithm using CSAPSO-BP neural network has higher accuracy, which verifies the effectiveness and superiority of the CSAPSO-BP neural network algorithm in the microwave detection of glass panel defects and the quantitative recognition of defects. The method overcomes the disadvantages of low efficiency, high labor cost and low detection accuracy of the human eye visual inspection method, and also avoids the harsh requirements of machine vision inspection on light source and camera shooting angle, which has certain guiding significance for the quantitative microwave identification of surface defects of non-metallic materials. However, due to the limited time and conditions of the study, the current research mainly focused on simulation and lacks certain experimental validation. Subsequently, it is necessary to build a microwave inspection test platform to verify the feasibility of the method. At the same time, the defects simulated in the paper are regular, while the defects detected on the actual production line are very complex, with different shapes of defects, which have a more complex impact on the detection effect; the detection of these natural defects needs to be focused upon in further research.

## Figures and Tables

**Figure 1 sensors-23-01097-f001:**
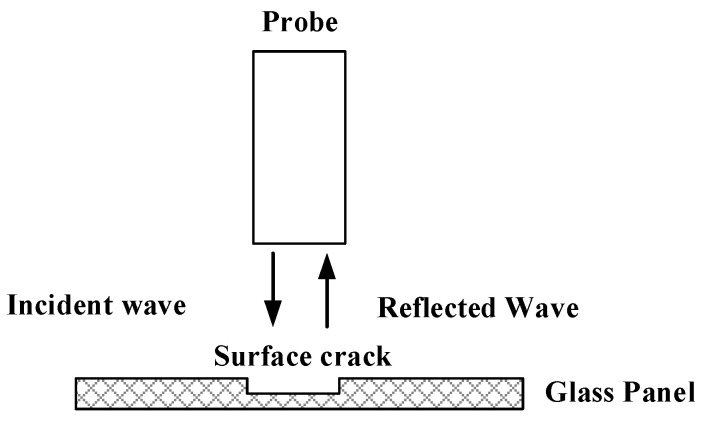
Microwave detection principle.

**Figure 2 sensors-23-01097-f002:**
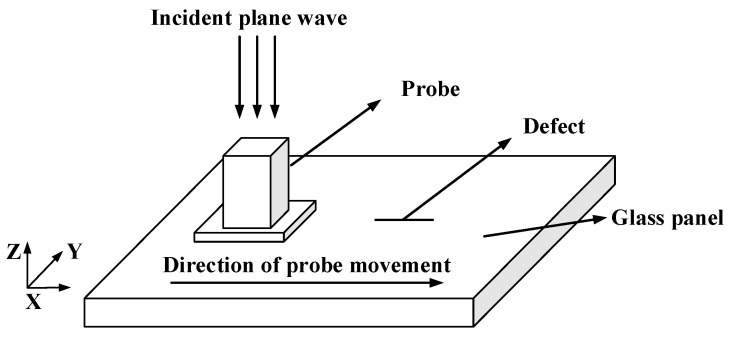
A 3D simulation model for the microwave inspection of the panel.

**Figure 3 sensors-23-01097-f003:**
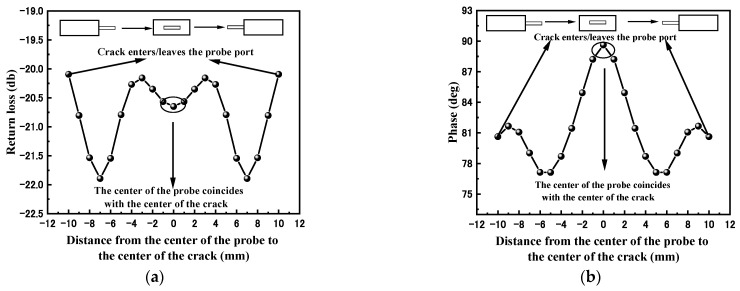
Echo signal characteristics of crack defects: (**a**) return loss variation curve; (**b**) echo phase variation curve.

**Figure 4 sensors-23-01097-f004:**
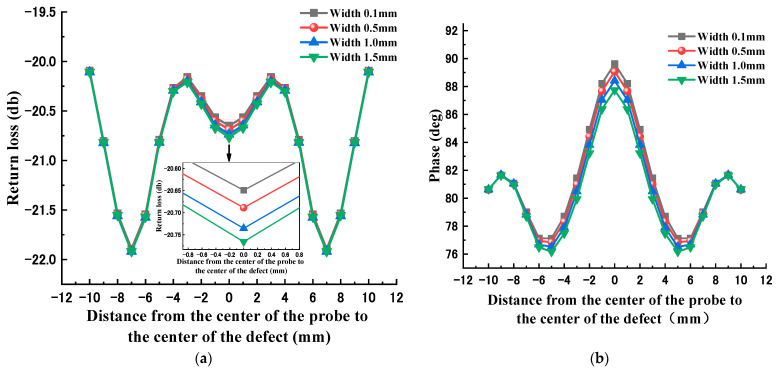
Echo signals of different width defects: (**a**) return loss variation curve; (**b**) echo phase variation curve.

**Figure 5 sensors-23-01097-f005:**
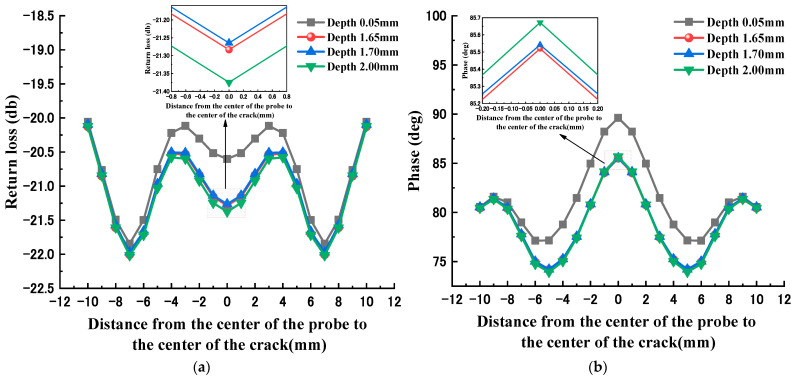
Echo signals of different depth defects: (**a**) return loss variation curve; (**b**) echo phase variation curve.

**Figure 6 sensors-23-01097-f006:**
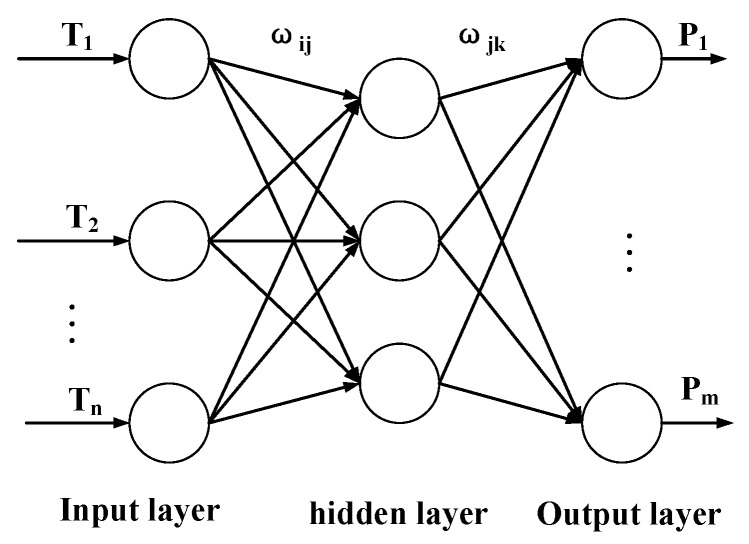
Structure of the BP neural network.

**Figure 7 sensors-23-01097-f007:**
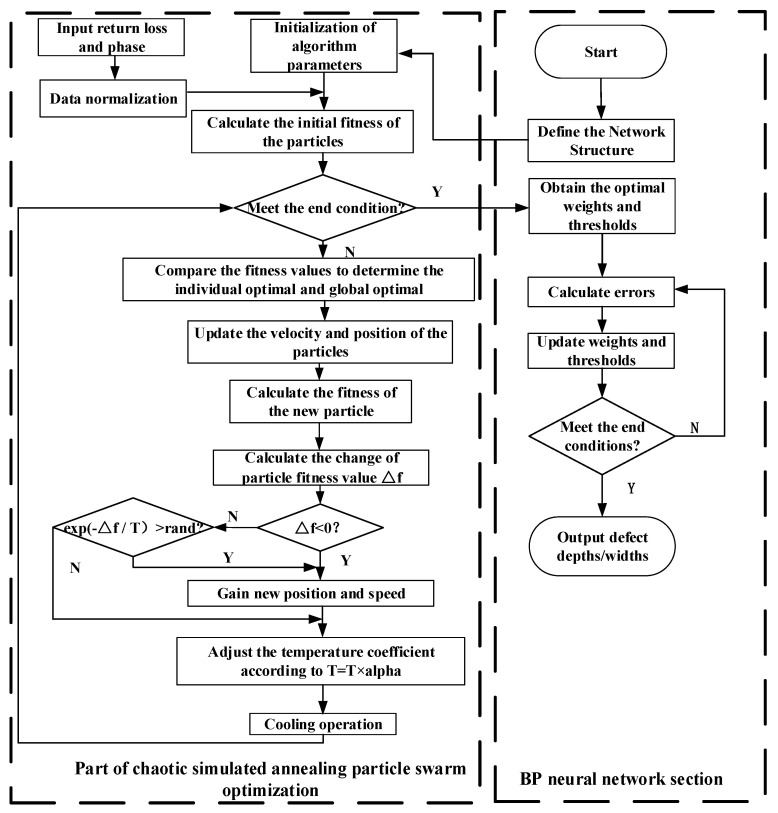
The CSAPSO-BP neural network algorithm.

**Figure 8 sensors-23-01097-f008:**
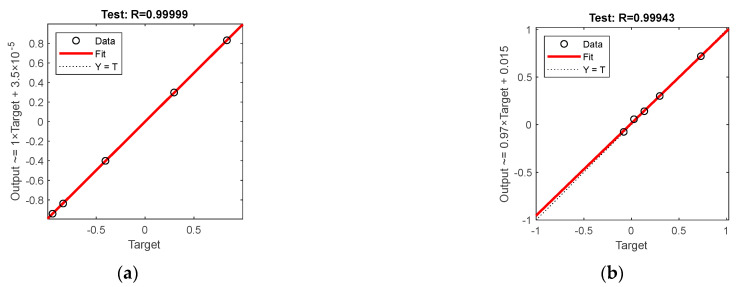
Test set R-factor for defect width prediction: (**a**) CSAPSO-BP algorithm; (**b**) PSO-BP algorithm.

**Figure 9 sensors-23-01097-f009:**
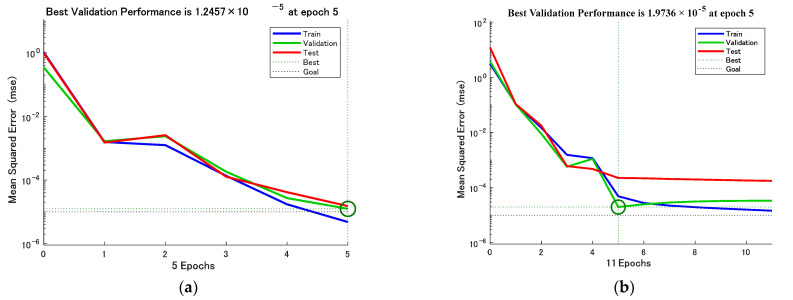
MSE variation curve for defect width prediction: (**a**) CSAPSO-BP algorithm; (**b**) PSO-BP algorithm.

**Figure 10 sensors-23-01097-f010:**
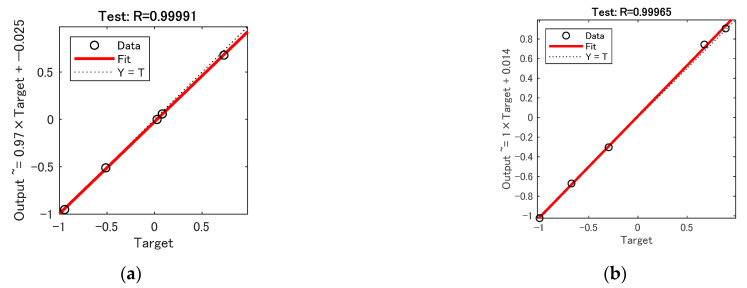
Test set R-factor for defect depth prediction: (**a**) CSAPSO-BP algorithm; (**b**) PSO-BP algorithm.

**Figure 11 sensors-23-01097-f011:**
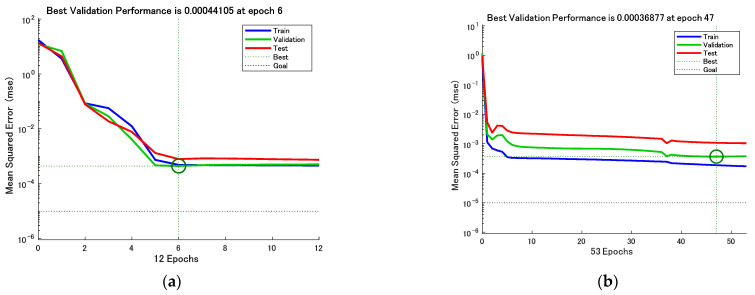
MSE variation curve for defect depth prediction: (**a**) CSAPSO-BP algorithm; (**b**) PSO-BP algorithm.

**Figure 12 sensors-23-01097-f012:**
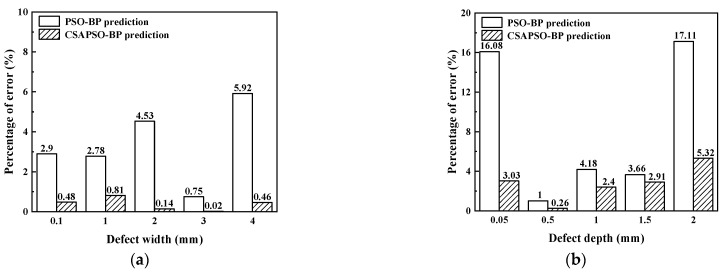
Echo signal characteristics of crack defects: (**a**) comparison of the prediction error of defect width; (**b**) comparison of the prediction error of defect depth.

**Table 1 sensors-23-01097-t001:** Comparison of defect width prediction results.

Serial Number	Actual Width Value/mm	PSO-BP Prediction	CSAPSO-BP Prediction
1	0.1	0.071008	0.095222
2	1.0	1.027841	1.008076
3	2.0	1.954659	1.998593
4	3.0	2.992504	3.000193
5	4.0	4.059224	3.995448

**Table 2 sensors-23-01097-t002:** Comparison of defect depth prediction results.

Serial Number	Actual Depth Value/mm	PSO-BP Prediction	CSAPSO-BP Prediction
1	0.05	0.210828	0.019655
2	0.50	0.489987	0.497413
3	1.00	1.041800	1.023970
4	1.50	1.463427	1.470877
5	2.00	2.171102	2.053180

## Data Availability

Not applicable.

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
