# Peer review of "Quantitative Identification Method for Glass Panel Defects Using Microwave Detection Based on the CSAPSO-BP Neural Network"

_sensors, 2023, doi:10.3390/s23031097_

Round 1

Reviewer 1 Report

In the manuscript, an inverse algorithm, i.e., CSAPSO-BP is proposed to inversely evaluate slots in glass panel via microwave reflectometry. There are a number of critical issues to be solved before further consideration: (1) why choose the waveguide with the aperture size 15.8mm*7.9mm? why is it related detection sensitivity? What is the wave mode (TE_X or TM_X) of the microwave generated by the waveguide? (2) regarding the relative permittivity, how is 4.82 determined? Note that the relative permittivity of most dielectric materials is complex value; (3) from Figure 3, it is indicated that the wave polarisation direction is perpendicular to the slot orientation. What about the case where polarisation is parallel to the slot orientation? (4) the raw testing signal, i.e., S11(frequency) is missing. Why choose 14.322GHz? (5) for training of BP, how many cases for training, test and validation? What about the R-value? How were the network parameters (from line 211-217) optimised? “After several tests” (line 213) is too bold; (6) the severe problem with the manuscript lies in the fact that the proposed algorithm along with the CST model is barely verified with experiments which in fact can be readily conducted. The MMW reflectometry system is simple whilst the regular-shaped slots can be easily fabricated on the glass panel. 

Reviewer 2 Report

Identifying defects on the surface of glass panels is an interesting topic. The authors proposed a new method of quantitative evaluation of microwave detection signals of glass panel defects, by combing CSAPSO with BP neural network. The manyscript is organizedd well, but I have the following questions.

On Section 2.2, a rectangular waveguide with a length of 15.8 and a width of 7.9 is choosed to improve the detection sensitivity. Why the authors choose this special size? Will a different size of waveguide affect the measurement results?

Lin 163, the authors combined the simulated annealing algorithm, and then the algorithm could accept some poor solution. This improved the search ability of the algorithm. I wonder that if the improvement will affect the speed of the global search? If so, which one is more important, the search ability or search speed?

In addition, there are some format mistakes in the reference information,

e.g. [3][6][8][14]... Please check them.

Round 2

Reviewer 1 Report

all my comments are addressed.